# The BCC7 Protein Contributes to the *Toxoplasma* Basal Pole by Interfacing between the MyoC Motor and the IMC Membrane Network

**DOI:** 10.3390/ijms23115995

**Published:** 2022-05-26

**Authors:** Luis Vigetti, Tatiana Labouré, Chloé Roumégous, Dominique Cannella, Bastien Touquet, Claudine Mayer, Yohann Couté, Karine Frénal, Isabelle Tardieux, Patricia Renesto

**Affiliations:** 1IAB, Team Biomechanics of Host-Apicomplexa Parasite, INSERM U1209, CNRS UMR5309, Grenoble Alpes University, 38700 Grenoble, France; luis.vigetti@univ-grenoble-alpes.fr (L.V.); tlaboure1998@gmail.com (T.L.); bastien.touquet@univ-grenoble-alpes.fr (B.T.); 2Université de Bordeaux, Team Microbiologie Fondamentale et Pathogénicité, CNRS UMR 5234, 33000 Bordeaux, France; roumegouschloe@gmail.com (C.R.); karine.frenal@u-bordeaux.fr (K.F.); 3IAB, Team Host-Pathogen Interactions & Immunity to Infection, INSERM U1209, CNRS UMR5309, Grenoble Alpes University, 38700 Grenoble, France; dominique.cannella@univ-grenoble-alpes.fr; 4Université Paris Cité, 75013 Paris, France; claudine.mayer@unistra.fr; 5ICube-UMR7357, CSTB, Centre de Recherche en Biomédecine de Strasbourg, 67084 Strasbourg, France; 6INSERM, University of Grenoble Alpes, CEA, UMR BioSanté U1292, CNRS, CEA, FR2048, 38000 Grenoble, France; yohann.coute@cea.fr

**Keywords:** Apicomplexa parasites, *Toxoplasma gondii*, cell biogenesis, cell polarity, cytoskeleton, basal pole complex, inner membrane complex, myosin C, myosin J, proteomics, expansion microscopy, STED microscopy

## Abstract

*T. gondii* is a eukaryotic parasite that has evolved a stage called tachyzoite which multiplies in host cells by producing two daughter cells internally. These nascent tachyzoites bud off their mother and repeat the division process until the expanding progenies escape to settle and multiply in other host cells. Over these intra- and extra-cellular phases, the tachyzoite maintains an essential apicobasal polarity that emerges through a unique bidirectional budding process of the elongating cells. This process requires the assembly of several molecular complexes that, at the nascent pole, encompass structural and myosin motor elements. To characterize a recently identified basal pole marker named BCC7 with respect to the posterior myosin J and myosin C motors, we used conventional biochemistry as well as advanced proteomic and in silico analysis in conjunction with live and super resolution microscopy of transgenic fluorescent tachyzoites. We document that BCC7 forms a ribbed ring below which myosin C motor entities distribute regularly. In addition, we identified—among 13 BCC7 putative partners—two novel and five known members of the inner membrane complex (IMC) family which ends at the apical side of the ring. Therefore, BCC7 could assist the stabilization of the IMC plaques and contribute to the parasite biomechanical properties.

## 1. Introduction

*Toxoplasma gondii* is an obligate intracellular protozoan parasite of a wide range of homeotherm animals worldwide. Following ingestion of host-to-host transmissible developmental stages called zoites, *T. gondii* reproduces asexually as either fast cycling and tissue-damaging tachyzoites, or as slow cycling and clinically quasi-silent bradyzoites in multiple cell lineages. Because of its ability to reversibly differentiate under tachyzoites and bradyzoites in deep and vital tissues of its hosts, *T. gondii* poses a significant health threat. In humans, ocular and cerebral toxoplasmosis are common recurrent and possibly life-threatening diseases of mostly—but not exclusively—immune-weakened individuals [1] while both also typifying congenital infection.

*T. gondii* belongs to the ancient phylum of Apicomplexa and shares with the other companion groups from the Alveolata infrakingdom, featuring a tri-lamellar pellicle outlining the cell [2] while a permanent apicobasal polarity also characterizes the *T. gondii* zoites. In addition to the plasma membrane bilayer, the pellicle encompasses the underneath inner membrane complex (IMC) [3] derived from flattened and fused vesicles identified as alveoli and made of a IMC and IMC-like protein’s repertoire, yet to complete. In close juxtaposition to this pellicle is the sub-pellicular alveolin-rich cytoskeleton which lies on a basket of 22 strikingly stable but flexible subpellicular microtubules [4,5]. While the microtubule network emerges apically from an atypical microtubule organizing center termed polar ring and extends over the two-thirds of the cell length, the IMC starts with an apical cone-shaped plate and follows with a patchwork of longitudinally aligned plates down to near the posterior edge [6,7]. Therefore, neither the apical nor the basal conic poles are outlined by the multilayer cortex. Importantly, all these structural membranous and cytoskeletal elements maintain complex molecular interactions between each other and with associated proteins, and jointly, they provide enhanced mechanical strength and flexibility to the elliptical zoite [8].

Beyond the long-lasting interest for the structural organization and functional contribution of the zoite apical complex to the motile and invasive behaviors of extracellular zoites [9,10], an increasing number of studies have started to highlight the specific composition and architecture of the basal pole and to model its progressive fabrication during the development of new zoites inside the host cells [7,11,12,13]. Zoites multiply by endodyogeny, a process through which two daughter cells assemble their cortex and cytoskeleton de novo inside an intact mother prior to cytokinesis. The progenies eventually bud out of the mother cell by recycling material from the mother plasma membrane while they complete cytokinesis. Successive rounds of division lead to typical rosettes of polarized tachyzoites whose basal poles are centripetally clustered around the maternal residual body through which they continue to connect to each other. Early studies have pointed out a spatial and temporal coincident biogenesis of the two poles in the daughter cells which subsequently elongate in an apical-to-basal direction [2,9]. A combination of advanced proteomics and cell imaging has recently allowed not only arguing for a bi-directional rather than unidirectional daughter cell budding process, but also discriminating four protein complexes with selective sub-cellular localization and assembly kinetics at the basal pole.

*T. gondii* expresses 11 myosins that distribute across five classes of the myosin superfamilies (i.e., class VI, XIV, XXII, XXIII, XXIV) [14], for which specific subcellular location and functional contribution have been resolved in large part. The basal pole of tachyzoites contains the class VI MyoJ and the class XIV MyoC which arises from differential splicing of the myosinB/C gene, and both delineate rings whereas the class XXIV MyoI was shown to accumulate behind at the residual body center of mature rosettes. Using gene invalidation strategies, MyoJ has been assigned a dispensable and actin-dependent role in the otherwise critical late constriction step of the daughter cell basal pole; a process that folds the basal cap post-cytokinesis whereas MyoI was involved in preserving communication between daughter cells, a prerequisite for cell division synchronicity [15].

Therefore, the assembly of basal pole components—myosin motors included—directs the biogenesis of the posterior cup in the two polarized daughter cells while it ensures proper rosette organization. Yet, in free tachyzoites, the structural arrangement of these components must be compatible with the generation of biological forces, in particular related to contraction and torque. As such, one can think of the peculiar twisting motion executed by the free tachyzoite to terminate invasion within an intracellular growth permissive vacuole [16]. Given that myosin motors can twirl actin in both left-handed and right-handed helical paths in mammalian systems, one can hypothesize that the invading tachyzoite generates an actomyosin-based torque to power the twisting motion. In the same line, torques are frequently observed when the free tachyzoites twirl upright against the 2D surface being posteriorly attached to it. Active exit of posteriorly constricted mature tachyzoites out of the host cells is also associated with a torque applied on the maternal residual body to free from it and individually undergo egress. Periodic contraction of the basal pole is also required for efficient traction force at the front to drive the typical tachyzoite helical motility [5]. Therefore, integrating how the myosin motors present at the basal pole could be efficiently anchored to scaffolding pieces of the basal pole—allowing the development of lateral forces or torques—would add to the understanding of the tachyzoite extracellular and intracellular life.

A proteomic analysis carried on a detergent-extracted subpellicular cytoskeleton fraction allowed identifying a 500 kDa protein encoded by TGGT1_311230 [17] which was subsequently selected in a Yeast two-hybrid (Y2H) screen with MORN1 as a bait [18] and therefore possibly involved in the shaping of the basal pole. We also identified this protein in a screen for *Toxoplasma* tachyzoite Histone Deacetylase 3 (HDAC3) substrate candidates and further invalidated its HDAC3 substrate potential, but we noticed the striking posterior localization of an epitope-tagged version of the protein. This in situ observation prompted us to map the protein position with respect to the posteriorly located myosins. In the course of our study, a BioID-based map of the basal complex (BC) proteome also picked up this protein as a basal complex component and named it as BCC7 [12,13]. Using a combination of biochemistry, co-immunoprecipitation coupled to mass spectrometry (MS)-based proteomics, and live and super resolution laser scanning microscopy, we now bring evidence for BCC7 positioned at the interface between a set of IMC proteins including two newly identified herein, and the MyoC motor. Similar to MyoC and the majority of IMCs studied so far, BCC7 was found dispensable for the tachyzoite lytic cycle, and in the same vein, the ultrastructural organization of the basal pole was not impacted in either resting or motile parasites lacking BCC7.

## 2. Results

### 2.1. BCC7, a Coccidia Restricted Protein with a Ring-Shaped Architecture Marking the Basal Pole of the T. gondii Tachyzoite

As indicated in ToxoDB [19], the TGGT1_311230 encodes a 4603 AA hypothetical protein of 494.8 kDa with a pI of 4.54 that harbors a N-terminal transmembrane domain. Identified as BCC7 in a previous proteomic analysis, this protein specifies the Coccidia within the Apicomplexa phylum but shows no structural homology or predicted functional association with any known proteins. However, the subcellular atlas of *T. gondii* tachyzoite proteins presumably assigned BCC7 to an IMC compartment based on a recent hyperLOPIT-based study [20]. We have now refined the amino acid sequence analysis by combining sequence composition analyses with Blast searches, which allowed highlighting 13 regions (Figure 1a). Each region was characterized by BLAST-hits, biased amino-acid composition, and degree of order/disorder. Two glutamic acid (E)-rich and a mix E/arginine (R)-rich (>25%) zones were identified. Four regions display significative BLAST hits for *T. gondii* sequences exclusively (residues 80–500) or for other Sarcocystidae including the *Neospora* and *Besnoitia* genera (residues 2530–3260 and 3260–3660). Interestingly, BCC7 appears intrinsically disordered except for a 730 aa in length domain (2530–3260) which is predicted as mainly ordered and might thus be folded. Bioinformatic analysis of BCC7 amino acid sequence also pointed out two VPV repeats (3340–3342; 3348–3350), the sequence of which is considered as a second repeat element that hallmarks alveolin proteins [21]. In addition, 129 phosphorylated amino-acid residues have been detected by phospho-proteomic analysis and the large majority (86%) of target serine in the C-terminal region along with four ubiquitin-binding sites that cluster at the extreme C-terminus of BCC7 (positions 4048, 4339, 4507, and 4461) (Appendix A).

To biochemically characterize this potentially membrane spanning and IMC-related BCC7 product, we conducted a series of sub-fractionation assays taking advantage of the tachyzoite line expressing a BCC7-HA-FLAG chimera at the endogenous locus. We first confirmed the Triton X-100 (TX-100) insoluble property of the BCC7 fusion protein in agreement with what has been described for the endogenous protein [17]. We then refined the extraction conditions to optimize protein recovery. Because proteins embedded in the IMC network to which BCC7 has been associated [20] were described as highly resistant to the detergents commonly used for membrane protein extraction [9,22,23], we analyzed the partitioning of BCC7-HA-FLAG under different solubilization conditions using Western blot and anti-HA antibodies. BCC7-HA-FLAG was found fully insoluble in the high salt BC500 lysis buffer containing NP40 (0.05% *v*/*v*) or in BC500 supplemented with DDM (0.1% *w*/*v*). In contrast, a significant amount of protein was recovered in the soluble fraction using the zwitterionic neutral Empigen BB (1% *v*/*v*) or sulfobetaine 3–14 (n-Tetradecyl-N,N-dimethyl-3-ammonio-1-propanesulfonate) (1% *w*/*v*) as well as the anionic N-lauryl sarcoside (0.1% *w*/*v*). It is worth noting that the Empigen-BB was the most efficient at solubilizing BCC7-HA-FLAG (Figure 1b). Although the migration in SDS-PAGE 4–12% gradient gels of BCC7 did not seem to depend on the detergent used and the solubility or insolubility status (Appendix A), we observed some variability in the migration pattern between experiments with products migrating faster but only one band was always observed arguing for a full-length product.

Next, in order to obtain in situ insights on the BCC7 distribution in tachyzoites, we generated parasites stably expressing various fluorescent- and epitope-tagged versions of BCC7 after insertion at the endogenous locus. The BCC7-mEmerald (mEm) expressing tachyzoites unambiguously delineated a ring that positioned near the edge of the basal pole of both intracellular and extracellular samples (Figure 1c). The same posterior location and circular shape were observed for tachyzoites expressing a BCC7-mCherry (mC) version or when using anti-HA antibodies to detect the chimeric BCC7-HA-FLAG protein (Appendix A). To document the dynamics of fluorescent BCC7 in cycling tachyzoites at a high spatial resolution, HFF cells were infected with BCC7-mEm tachyzoites which provided the brightest direct fluorescent BCC7 signal. Following a 30 min period of contact between freshly egressed tachyzoites and HFF monolayers, the non-internalized parasites were removed and the development of the internalized ones was further monitored. Within two to four hours post invasion, we observed newly synthetized BCC7-mEm pools at the very front of the tachyzoite where the IMC emerges with a symmetrical pair or triad dot pattern. With time, BCC7-mEm pools were detected slightly more posteriorly, shaping peripheral successive dots or elongated spots (e.g., possibly merging dots) along the cell edge nearby the IMC, a pattern suggesting trafficking features (Figure 1d, see arrows). This multiple dot-like localization recalls the one reported for the canonical marker of the posterior cup at the extreme basis of mature tachyzoites, namely centrin2 (CEN2) which was also detected at the polar ring and peripheral annuli during the interphase [9,11]. Similar to CEN2 but also to MORN1, another canonical marker of the basal complex, we occasionally detected BCC7-mEm in close juxtaposition of the nucleus apical side but we did not see signals nearby the duplicated centriole as typically observed for CEN2 [11]. When each budding daughter cell has then synchronously extended their IMC posteriorly, BCC7-mEm pools were detected as forming individual rings at the nascent basal pole of each progeny similarly to what is known for MORN1, and these rings progressively gained in signal intensity suggesting addition of material (Figure 1c). To reach better resolution of BCC7-mEm in intracellular tachyzoites, we used STimulated Emission Depletion (STED) microscopy and confirmed the travelling of BCC7 pools along the peripheral IMC network at a 25 nm resolution, as well as their integration in the basal pole of each daughter cell (Figure 1e). Next, to monitor in live the dynamics of the BCC7 trafficking during endodyogeny, we video-recorded BCC7-mEm signal with respect to the tachyzoite nucleus over several rounds of cell cycle, which highlights the BCC7 apicobasal trafficking. Live imaging also indicated that the two BCC7-mEm de novo pools which assemble into rings at the cytoskeleton base of the budding daughters can integrate some maternal BCC7 product at the latest step, thereby suggesting that both de novo synthesis and recycling could contribute to the BCC7 found at the basal cap of mature tachyzoites (Figure 1d and Appendix A).

### 2.2. Super Resolution Microscopy Highlights the Close Proximity between MyoC and BCC7

To assess the position of BCC7 with respect to actomyosin motors known to assemble as a ring at the basal complex, namely MyoC and MyoJ, we imaged tachyzoites co-expressing either BCC7 HA-Flag and MyoC-YFP, BCC7-mC and MyoC-YFP, or BCC7-HA-Flag and MyoJ-mC. Using widefield fluorescence microscopy coupled to structured illumination through the Apotome module, we observed for both extracellular and intracellular tachyzoites that the MyoC-YFP signals were overlapping those from either the immuno-fluorescently labeled BCC7-HA-Flag or the fluorescent BCC7-mC products (Figure 2a and Appendix A). By contrast, the 4.54 pixel size of the camera allowed detecting two apposed but separate fluorescent signals when BCC7-HA-Flag was co-detected with MyoJ-mC (Appendix A). To better map the relative position of the BCC7 and MyoC proteins, we embedded samples in swellable hydrogels and processed them for an isotropic mechanical expansion following recently developed protocols of the so-called expansion microscopy (proExM) [25]. Under these isotropic expansion conditions (which allowed a linear estimated 4X expansion), and using a 6.5 μm pixel size CMOS Prime camera, we increased the resolution up to about 70 nm and were indeed able to discriminate the two molecular species, hence to position MyoC just below the BCC7 ring as attested by the more central nucleus location (Figure 2b). Moreover, when we applied the Airyscan technique to improve signal to noise ration and resolution on the proExM gel sample, we gained descriptive information with a regular distribution of MyoC elements around the continuous circular BCC7 signal (Figure 2c). This in-depth imaging analysis argues for a close physical partnership between the motor and BCC7. Of note, well-defined rings of MyoC were detected in the assembling daughter cells coinciding with BCC7 signals, in line with a spatially co-assembly process (Figure 2a bottom panel, Figure 2d).

### 2.3. The BCC7 Protein Is Dispensable for the Tachyzoite Extracellular and Intracellular Lifestyle

Considering the large size and the position of BCC7 relative to the MyoJ and MyoC motors, we wanted to check whether mutants devoid of BCC7 would have impaired actomyosin-dependent critical behaviors (e.g., motile, replicative, and invasive abilities). To this end, we generated an inducible mutant of BCC7 using the auxin-inducible degron (AID) system proved to drive the specific and conditional depletion of a target protein in presence of auxin (indoleacetic acid, IAA) [26] (Appendix A). Indeed, when the HFF monolayer was loaded with BCC7-mAID-3HA mutants and further incubated with IAA for 40 h, the BCC7-HA tag became undetectable under IF conditions, arguing for an efficient depletion of BCC7. It is noteworthy that the BCC7 altered neither the daughter cell assembly process including its polar orientation nor the development of rosette-like structures (Figure 3a). Using real time confocal microscopy, we next monitored the motile behavior of both wild-type and mutant tachyzoite populations. The tachyzoites for which BCC7 was undetectable remained typically organized in rosettes within a parasitophorous vacuole which dismantled upon active egress of the BCC7-depleted progenies, this with indistinguishable features from the sequences recorded with BCC7 expressing tachyzoites. Similarly, newly egressed BCC7 mutants displayed typical wild-type circular and helical gliding modes of motility as well as twirling behavior (Figure 3b, Appendix A). The absence of apparent defects of motile, replicative, and invasive behaviors in vitro was recapitulated by the similar surface of HFF destruction as a result of the wild-type and mutant tachyzoite ‘lytic cycle’ using typical plaque assays over a 7-day period (Figure 3c). To then check for any ultrastructural change at the cell surface in particular at the basal pole that would be driven by the lack of BCC7, we applied scanning electron microscopy on wild-type and mutant tachyzoite samples. Focusing on resting (e.g., conoid retracted) but also gliding tachyzoites (e.g., as attested by both the apical flexure [5] and the trail left behind), we could appreciate the significant stretching of the basal pole that elongates as a thin elastic cone in gliding tachyzoites when compared to no gliding ones, but this occurred regardless of BCC7 expression (Figure 3d,e). Collectively, despite the large size of BCC7, these results argue for a dispensable BCC7 contribution to the biogenesis of a functional basal pole and consequently for the perpetuation of tachyzoites in vitro, in line with the fitness score retrieved in a genome-wide knock out CRISPR screen [27].

### 2.4. Identification of Putative BCC7-Interacting Partners by Co-Immunoprecipitation Coupled to Mass Spectrometry Analysis

Considering the highest yield of BCC7-HA-FLAG recovered with Empigen BB and the capacity of this zwitterionic detergent to preserve antigenicity of solubilized proteins [28], the latter was subsequently used for co-immunoprecipitation experiments. To obtain a BCC7-HA-FLAG enriched extract, a two-step subcellular fractionation method based on sequential lysis was applied. A first lysis step was carried with the non-ionic detergent TX-100 (1%, *v*/*v*), and allowed the release of many cytosolic proteins including a significant fraction of the MyoA motor known to be mainly housed within the pellicle space [29]. As expected, these conditions did not allow solubilizing BCC7-HA-FLAG (Figure 1b and Appendix A), and we applied a second extraction step by adding Empigen BB (at the concentration defined previously, see Figure 1b) to the BBC7-HA-FLAG positive TX-100 insoluble fraction. After clarification of the lysate, the soluble extract was enriched in BCC7-HA-FLAG and could therefore be subjected to immunoprecipitation with anti-*FLAG*^®^ M2 antibodies covalently linked to agarose beads. While the solubilized BCC7-HA-FLAG pool was entirely immobilized on the anti-FLAG agarose under the experimental conditions, it was subsequently recovered in the two first eluate fractions collected after competitive elution with FLAG peptides (Appendix A). Having set the right conditions to capture BCC7-HA-FLAG from tachyzoite extracts, we could upscale the assays in order to identify putative partners of BCC7-HA-FLAG. To this end, three replicates of this co-immunoprecipitation procedure were performed using in parallel the BCC7-HA-FLAG-expressing *T. gondii* line and the parental one (RHΔku80) which served as control. For both types, the eluate samples were pooled and processed for MS-based quantitative proteomics. Comparative statistical analysis allowed ranking of 13 *Toxoplasma* proteins that were significantly enriched (four-fold) in the BCC7-HA-FLAG eluates as compared to controls, among which several IMC proteins (Table 1). Aside from the already identified IMC7, IMC10, IMC12, IMC18, and IMC24, three other proteins annotated as hypothetical proteins and encoded by TGGT1_230160 (Partner 8), TGGT1_236950 (Partner 9), and TGGT1_231160 (Partner 13 untagged) have been captured in our screen and have previously been putatively assigned to the IMC subcellular compartment after their endogenous tagging [20]. The list also includes the SAG-Related Sequence SRS29A protein (TGGT1_233450), a surface-exposed protein from the SAG glycoprotein superfamily [30] and TgZFP2 (TGGT1_212260; Partner 6). TgZFP2 is zinc-finger protein with features of a cytoskeleton-associated protein which has been reported essential for parasite survival [31]. Two additional putative BCC7-HA-FLAG partners were identified as a 155 kDa protein of unknown function and localization (TGGT1_230940; Partner 3), and a nuclear FUSE-binding protein 2 (TGGT1_216670; Partner 7), both proteins being predicted a high contribution to parasite fitness [27]. The last BCC7-HA-FLAG partner candidate is one hypothetical protein that has not yet been assigned to any sub-cellular compartment (TGGT1_315610; Partner 12).

### 2.5. A Majority of IMC Proteins, Including Two New Ones Were Identified in the BCC7-Binding Protein Screen 

To characterize the putative BCC7-HA-FLAG partners in situ, we engineered a series of lines that co-express BCC7-mEm and a mCherry(mC)- or myc-tagged given target in the RHΔku80 background strain. As observed in Figure 4, the five IMC-mC fusion proteins (e.g., IMC7, IMC10, IMC12, IMC18, and IMC24) showed a typical IMC peripheric sub-membranous distribution [32,34], thereby attesting to the functionality of the given fusion protein. It is worth noting that, while these IMC-mC fusions ran up to just above the BCC7 ring, only the IMC10 was detected into the growing cytoskeletons of nascent daughter cells (Figure 4, Appendix A). Outside from the IMCs, the TgZFP2 zinc finger protein (Partner 6) reported to critically assist daughter cell budding through a partnership with the cytoskeleton [33], shows a punctate cytoplasmic distribution intracellular tachyzoites (Appendix A). Notably, previous attempts to identify putative molecular partners of this protein have remained unsuccessful [33]. In contrast with a predicted nuclear localization, the protein encoded by TGGT1_216670 (Partner 7) and annotated as FUSE-binding protein 2/KH-type splicing regulatory protein was observed dispersed in the tachyzoite cytoplasm when expressed as a chimera with the mCherry protein (Appendix A). Partner 3 is a protein of unknown function with no current assignment to any compartment, and while we failed to generate a mCherry fusion version of the protein, we succeeded in producing tachyzoites stably expressing the myc-tagged Partner 3. Using anti-myc antibodies, we detected a cytoplasmic localization with a patchy distribution (Appendix A). Finally, the Partner 12 (TGGT1_315610) without assigned sub-localization, was found to be nucleus associated when tachyzoites expressed either mCherry or HA tags (Appendix A). We remained unsuccessful at generating a transgenic parasite line expressing the endogenously SRS29A-tagged SAG-Related Sequence protein which, given its high immunogenicity [30] and predicted localization [20], is likely to be exposed at the tachyzoite plasma membrane.

Specific attention has been given to two other BCC7-putative partners, namely Partners 8 (TGGT1_230160) and 9 (TGGT1_236950) that are conserved across the phylum of Apicomplexa and have been classified as dispensable [27]. Both displayed an IMC-like localization when expressed as mCherry fluorescent versions (Figure 5a,b), in agreement with their predictive assignment as IMC proteins [20]. To the best of our knowledge, the last IMC protein identified in *T. gondii* is IMC34 [35]. Accordingly, we propose to name Partner 8 and Partner 9 as IMC35 and IMC36, respectively. When image resolution reached 25 nm owing to STED microscopy, we could visualize not only the relative position of the IMC35 with respect to the underlying microtubule regular network with the IMC35 grid-like signal distribution, but also the very close proximity between the BCC7-mEm ring and IMC35 (Figure 5c). Remarkably, with this resolution, the BCC7 ring appears ribbed with grooves while small claws of BCC7 material could be seen facing up the IMCs, here exemplified with the IMC35 and IMC18, and these could possibly contribute to the firm attachment of the IMC basal plaques (Figure 5c,d). Finally, to gain some insights on the dynamics of these novel IMC proteins, we monitored their expression pattern using high-resolution live-cell imaging. We observed that expression and localization of both IMC35 and IMC36 are cell cycle regulated, which fits with the typical transcriptional [36] and protein expression profiles [3,37,38] of several IMCs. Periodic fluctuations in the fluorescent signal intensity of both IMC35-mC and IMC36-mC-expressing tachyzoites were observed during intracellular multiplication with a progressive increase in expression and in pellicle association, along with the late steps of daughter cell formation. Indeed, no labeling of budding progeny that would sign for an early integration of the protein into the nascent cytoskeleton of the daughter cells has been detected. In addition, the IMC35-mC and IMC36-mC kept a strong fluorescent IMC pattern in extracellular parasites as observed post egress using live and static microscopy (Appendix A; Figure 5).

## 3. Discussion

*Toxoplasma* zoites acquire a typical polar architecture as early as it assembles within a mother cell while undergoing a peculiar mode of division referred as to endodyogeny. Most of our knowledge concerning how polar ends form in nascent progenies has been gained on the *T. gondii* tachyzoite developmental stage taking advantage of both its fast-cycling properties in most mammalian host cells and its genetic tractability. Indeed, pioneering studies have uncovered that the daughter cells emerge with the simultaneous de novo building of the apical and basal complexes nearby the duplicated centriole at the apical side of the mother nucleus [9,11]. In this study, we focused on the basal complex known as a dynamic structure made of several ring-like protein arrangements that caps posteriorly, thereby limiting the elongating daughter cells. An essential myosin-J dependent constriction of the basal complex allows shaping of the definitive basal compartment of the mature progenies [15]. While significant progress has been made in the inventory and in situ relative mapping of main molecular elements of the basal complex, a comprehensive understanding of the basal pole functional properties is still uncompleted. How do the endodyogeny mechanisms lead to the emergence of progenies of roughly the same size from generation to generation inside the expanding intracellular vacuole remain mysterious. Moreover, free from the subpellicular microtubule scaffold but enriched in actin filaments and myosin motors, the basal pole of mature tachyzoites has shown great elasticity, a property that supports the periodic extension and contraction cycles of the basal pole requested to glide out of the cell and throughout extracellular matrices [5]. 

With the imaging at a few tens of nanometers of the BCC7 protein recently identified [12,13] but not yet resolved in situ, we now document that this remarkably large protein of about 500 kDa organizes as a ribbed ring around the conic basal pole. Interestingly, one can distinguish some kind of claws that extend from the ring-like base and could promote better stabilization of the molecular pieces lying on top for optimal biomechanical coordination. On the other hand, super-resolution and live imaging brought evidence for the BCC7 apicobasal trafficking with clusters of proteins running peripherally at the pellicle vicinity. Finally, the conjunction of biochemical assays, live and static imaging has pinpointed the close apposition of the IMCs—in particular, IMC35 with BCC7—but also the quite close partnership with MyoC seen in this study as discrete entities distributed just below the BCC7 ring. Of note, similar to BCC7, MyoC was also reported to partition with the subpellicular cytoskeleton TX-100-insoluble fraction [17] and to display an overlapping transcriptional profile with a peak of expression during mitosis (M)/cytokinesis (C) phases of the cell cycle [36]. MyoC has been reported to co-localize with MORN1, both forming rings at the base of nascent daughter cells [39], and BCC7 was picked up both in the Y2H screen with MORN1 as a bait [18] and a BioID CEN2 screen [13]. Future investigations should clarify the in situ partnership between the three players. 

BCC7 is a much larger protein that the canonical markers of the basal pole such as CEN2 and MORN1, and despite an accurate in silico analysis, no evidence for predictive functional domains was found. However, specific features and signature motifs could assist future characterization. First, given the enrichment of the BCC7 amino acid sequence in glutamic acid, the most common disorder-promoting amino acid, one can speculate on the BCC7 capacity to interact with several unrelated binding partners and to consequently impact on the 3D architecture [40]. Similarly, intermolecular partnerships could be favored through the large E and E/R-rich sequence repeats often involved in the regulation of protein–protein interactions [40,41]. The two BCC7 VPV repeats argue for a structural proximity with alveolin proteins [21,23]. These observations fit with a bioinformatic analysis, showing that 21% of the putative pellicular proteins contain highly repetitive regions with strong amino-acid biases for particular residues including glutamic acid [42]. Another striking BCC7 sequence feature is the high number of phosphorylation/dephosphorylation sites (Appendix A). As described for glutamic acid enrichment, protein phosphorylation/dephosphorylation events are major regulatory processes for protein activities by affecting protein structure, stability, subcellular localization, as well as interaction with other biomolecules, and not surprisingly, these post-translational modifications (PTMs) critically control many steps along the tachyzoite lytic cycle [43]. Another BCC7 predicted PTM is ubiquitination, which would be restricted to the BCC7 extreme C-terminus while it usually characterizes cytoskeletal and IMC proteins of *T. gondii*, many of which contribute to the cell division and host cell invasion processes [44]. By targeting proteins to the proteasome, polyubiquitination directs their proteolytic degradation. Notably, turnover of proteins can also depend on additional degradation signals in particular the phosphorylation of PEST sequences rich in proline (P), glutamic acid (E), serine (S), and threonine (T) residues [40]. Interestingly, three PEST motifs were detected but these are located at the BCC7 N-terminus. Collectively, the large repertoire of predictive PTMs on BCC7 amino-acid residues suggests that ubiquitination and phosphorylation could interplay to fine tune BCC7 properties, as already proposed for the complex regulatory mechanisms underlying tachyzoite endodyogeny [44]. However, that BCC7 was found dispensable in vitro for *T. gondii* tachyzoite perpetuation based on plaque formation capacity was surprising. A detailed study by Gubbels et al. published in the course of our manuscript evaluation [12] points out the subtle effect of BCC7 loss following gene knock out strategy on the rosette organization of the progeny within vacuole which appears to be less consistent than in wild type, and which do not impact on the general in vitro fitness score. This defect suggests that BCC7 could assist in the formation and maintenance of the cytoplasmic bridge between progeny after division within the parasitophorous vacuole which would translate into a rosette disorganization. It is also worth noting that other BC components identified in the Gubbels et al. study were found to separately affect the parasite alignment in rosette to various extents, indicating that *T. gondii* has likely evolved a versatile multi-component process that introduces molecular plasticity in the shaping of mature cell upon the final step of division.

In conclusion, BCC7 is the largest protein identified so far in the *T. gondii* basal pole of the nascent and mature tachyzoites that we found to closely interface between the IMC network and the myosin C motor which has not been assigned a highly specific functional contribution so far. Future investigations regarding the structural determinants of BCC7 oligomerization that result in the formation of ordered ring structures could now be envisaged using cryoEM for example. A better knowledge of the spatial architecture of BCC7 in the context of the whole set of structural players—also including the actin cytoskeleton and the myosin motors—should improve our understanding of the structural and functional peculiarities of the early designed basal pole cell compartment of *T. gondii* tachyzoite.

## 4. Materials and Methods

### 4.1. Parasites and Host Cells

Unless specified all reagents used for cell culture were from Gibco-Life Technologies. *T. gondii* strains (RH and RH-Δ*ku80* and derivatives) were propagated in human foreskin fibroblasts (HFFs, ATCC CCL-171) cultured at 37 °C under 5% CO_2_ atmosphere in Dulbecco’s modified Eagle’s medium (DMEM) supplemented with 10% heat-inactivated fetal bovine serum (Invitrogen), 10 mM HEPES buffer pH 7.2, 2 mM L-Glutamine, 100 U/mL penicillin, and 50 µg/mL streptomycin.

### 4.2. Generation of Transgenic Parasites Expressing Endogenously Tagged Proteins

Endogenous gene tagging was carried out using primers and plasmids listed Appendix A. Briefly, the C-terminus coding sequences of genes of interest were amplified with specific forward and reverse primers and using RHΔ*ku80 T. gondii* strain genomic DNA as template. The resulting PCR products were inserted into various pLIC vectors following the ligation-independent cloning protocol [45]. The BCC7-mEmerald fusion protein was generated via overlapping PCRs of TGGT1-311230 and of EmGFP-SUN2, amplified from genomic DNA and synthetic TUB8-EmGFP-SUN2 gene (Genscript), respectively. The reverse primer of the whole amplicon was engineered to add a STOP codon downstream the Emerald coding sequence. The correctness of the fusion was confirmed by DNA sequencing and the PCR product was incorporated in pLIC-HA-FLAG-HX vector (Gift of M.-A. Hakimi). For transfection, 10^7^ of freshly egressed RHΔku80 parasites washed then re-suspended in 800 μL of cytomix (10 mM K2HPO4, 10 mM KH2PO4, 120 mM KCl, 0.15 mM CaCl2, 5 mM MgCl2, 25 mM HEPES, 2 mM EGTA) were electroporated with 10–15 μg DNA in 2 mm gap cuvettes and using a BTX ECM 630 machine (Harvard Apparatus) at 1100 V, 25 Ω, and 25 F. Transgenic parasite populations were enriched using mycophenolic acid (25 μg/mL; Sigma-Aldrich) and xanthine (50 μg/mL; Sigma-Aldrich) for HXGPRT selection or pyrimethamine (3 μM) for DHFR selection, and individual clones were isolated by limiting dilution in 96-well plates. 

### 4.3. Bioinformatic Analyses

Sequence of BCC7 was retrieved from ToxoDB [19]. Several sequence analyses tools were combined to Blast searches. A transmembrane helix has been predicted by TMHMM-2.0 webserver [46]. PEST motif identification was performed using the EMBOSS program ePESTfind tool using the standard setting. Three potential PEST motifs (224–258, 313–347, 427–475)—with PEST scores of 6.2, 8.9, and 5.5 respectively—have been identified [47]. Composition in amino acids of each region has been recovered with the ProtParam tool of Expasy [48]. Ordered and disordered regions have been predicted using the FoldIndex webserver [49]. Predicted phosphorylation sites were retrieved from ToxoDB documented based on the phosphoproteomic studies on *Toxoplasma* and *Plasmodium* zoites [50].

### 4.4. Auxin-Inducible Degron Plasmid Construction 

The BCC7-mAID-3HA cell line was generated using the pU6-Cas9-sgRNA-BCC7 plasmid, which targets the 3′ end of the gene, near the stop codon, for the double strand break and a PCR product generated with the Q5 High-Fidelity DNA Polymerase (NEB) using the pTUB1:YFP-mAID-3HA, DHFR-TS:HXGPRT vector as template [51] and the primers 1875/1876 to repair (Appendix A). The pU6-Cas9-sgRNA-BCC7 plasmid was generated by annealing primers 1873 and 1874 and ligating them in the BsaI-digested pU6-Universal vector (gift from Sebastian Lourido [52]). The PCR product containing 33 bp homology to BCC7 and the pU6-Cas9-sgRNA-BCC7 were co-transfected in the RH TIR1-3FLAG cell line allowing integration and selection using mycophenolic acid and xanthine as described in Section 4.2. Downregulation of BCC7 was performed by incubation with a final concentration of 500 μM of IAA (Sigma).

### 4.5. Plaque Assays 

Freshly egressed parasites were collected, counted, and seeded to invade confluent HFF monolayers grown on 24-well plates. Parasites were allowed to invade overnight before treatment with 500 µM IAA or vehicle for 7 days. Monolayers were later methanol fixed and 0.1% crystal violet (Sigma) stained for plaque formation assessment. 

### 4.6. BCC7 Solubilization Assays

RHΔ*ku80_BCC7-HA-FLAG* parasites were pelleted (900× *g* for 7 min) and washed in cold PBS supplemented with complete protease inhibitor cocktail EDTA free (Roche Diagnostics) and 1 mM Phenylmethylsulfonyl fluoride (PMSF). The pelleted parasites were resuspended at a density of ∼1 × 10^9^ cell per mL of ice-cold lysis buffer (20 mM Tris-HCl pH 7.9–500 mM KCl—1.5 mM MgCl_2_—0.2 mM EDTA—0.05% NP-40—20% glycerol containing 1 mM PMSF and protease inhibitor cocktail). Parasites were then disrupted by 15 strokes in a Dounce homogenizer and 100 µL aliquots of the cell homogenate were distributed in Eppendorf tubes in presence of various detergents (BC500: control lysis buffer containing NP40 (0.05% *v*/*v*); BC500 supplemented with *n*-Dodecyl-β-D-maltopyranoside (0.1% *w*/*v*); Empigen BB (1% *v*/*v*); N-lauryl sarcoside (0.1% *w*/*v*); sulfobetaine 3–14 (n-Tetradecyl-N,N-dimethyl-3-ammonio-1-propanesulfonate) (1% *w*/*v*); Triton X-100 (0.1% *v*/*v*). Following a 90 min incubation on a rotating shaker, soluble and insoluble fractions were separated by centrifugation (12,000× *g* for 30 min, 4 °C) and further analyzed by Western blot. All of the purification steps have been carried out at 4 °C.

### 4.7. Immunoprecipitation Procedure

Freshly egressed parasites collected from 20 T180 flasks (1.3–1.5 × 10^10^ tachyzoites) were washed as described above and resuspended in 5 mL PBS supplemented with TX-100 1% (*v*/*v*). All lysis steps were performed at 4 °C in presence of antiproteases (Complete, Roche) and 1 mM PMSF. A mechanical lysis step was performed with Dounce homogenization of parasites followed by 1 h incubation on rotating shaker. The insoluble fraction collected after centrifugation (16,000× *g* for 20 min) was resuspended in lysis buffer (1 × 10^9^ tachyzoites/mL) supplemented with 1% Empigen BB (Sigma-Aldrich) and 1 µL Benzonase (Sigma-Aldrich). This BCC7 enriched fraction was further submitted to a second step of mechanical Dounce homogenization and 1 h incubation on rotating shaker. The soluble lysate obtained after centrifugation (20,000× *g* for 45 min) was diluted half in lysis buffer without detergent and incubated with 1 mL anti-FLAG M2 affinity beads (Sigma-Aldrich). After 1 h incubation, unbound material was washed away from the beads with 30 mL PBS and elution was carried out with 150 mg/mL FLAG peptide. The presence of HA-FLAG-tagged proteins in the elution fractions was assessed by immunoblot analysis after SDS-PAGE. Experiments were conducted in triplicate with both RHΔku80_BCC7-HA-FLAG, and wild-type parasites were used as controls.

### 4.8. Western Blots

The samples were resolved by SDS-PAGE (NuPAGE^TM^ 4–12% Bis-Tris Gel; Thermo Fisher Scientific) at 130 V using NuPAGE^TM^ MOPS SDS running buffer before transfer on PVDF membranes (2 h, 100 V) using NuPAGE^TM^ transfer buffer supplemented with 0.05% (*w*/*v*) SDS in order to ease the transfer of large proteins such as BCC7. Following a saturation step with PBS—Tween 20 (0.2% *v*/*v*)—nonfat milk (5% *w*/*v*), membranes were incubated with a rabbit monoclonal anti-HA antibody (C29F4; 1:1000; Cell Signaling) and a rabbit secondary antibody coupled to peroxidase (1:10,000; Jackson ImmunoResearch, Baltimore, PA, USA). After extensive washes with PBS-Tween, Western blot were revealed using Clarity Western ECL substrate (Bio-Rad) using ChemiDocTM XRS+ with image LabTM Software.

### 4.9. Mass Spectrometry-Based Quantitative Proteomic Analyses

Eluted proteins were solubilized in Laemmli buffer and stacked in the top of a 4–12% NuPAGE gel (Invitrogen). After staining with R-250 Coomassie Blue (Biorad), proteins were digested in gel using trypsin (modified, sequencing purity, Promega), as previously described [53]. The resulting peptides were analyzed by online nanoliquid chromatography coupled to MS/MS (Ultimate 3000 RSLCnano and Q-Exactive HF, Thermo Fisher Scientific) using a 140 min gradient. For this purpose, the peptides were sampled on a precolumn (300 μm particle size × 5 mm length PepMap C18, Thermo Scientific) and further separated in a 75 μm × 250 mm C18 column (Reprosil-Pur 120 C18-AQ, 1.9 μm, Dr. Maisch). The MS and MS/MS data were acquired by Xcalibur (Thermo Fisher Scientific). Peptides and proteins were identified by Mascot (version 2.7.0.1, Matrix Science) through concomitant searches against the *T. gondii* database (ME49 taxonomy, v.30, downloaded from ToxoDB), the Uniprot database (*Homo sapiens* taxonomy, October 2020-10 version), and a homemade database containing the sequences of classical contaminant proteins found in proteomic analyses (bovine albumin, keratins, trypsin, etc.). Trypsin/P was chosen as the enzyme for protein digestion and two missed cleavages were allowed. Precursor and fragment mass error tolerances were set at respectively at 10 and 20 ppm. Peptide modifications allowed during the search were: carbamidomethyl (C, fixed), acetyl (Protein N-term, variable), and oxidation (M, variable). The Proline software [54] was used for the compilation, grouping, and filtering of the results (conservation of rank 1 peptides, peptide length ≥ 6, PSM score ≥ 25, false discovery rate of peptide-spectrum-match identifications <1% as calculated on peptide-spectrum-match scores by employing the reverse database strategy, and a minimum of 1 specific peptide per identified protein group. Proline was then used to perform a MS1 quantification of the identified protein groups, based on razor and specific peptides.

Statistical analysis was then performed using the ProStaR software [55]. Proteins identified in the contaminant database, proteins identified by MS/MS in less than two replicates of BCC7-HA-FLAG eluates, and proteins detected in less than three replicates of one condition were discarded. After log2 transformation, abundance values were normalized by median centering, before missing value imputation (salsa algorithm for partially observed values in the condition and DetQuantile algorithm for totally absent values in the condition). Statistical testing was then conducted using limma, whereby differentially expressed proteins were sorted out using a log2(fold change) cut-off of 2 and a *p*-value cut-off of 0.05, leading to FDR below 2% according to the adjusted Benjamini–Hochberg (abh) estimator.

### 4.10. Immunofluorescence Microscopy

Intracellular (in HFF cells) or extracellular tachyzoites were fixed for 20 min at room temperature with 4% paraformaldehyde (PFA) diluted in PHEM pH 7.5 and dH_2_O then treated 10 min with 50 mM NH_4_Cl to quench free aldehyde groups before TX-100 permeabilization (0.1% *v*/*v*; 10 min). Following a 20 min non-specific site saturation step in PBS-BSA 2% (*w*/*v*) the cells were incubated with antibodies (Appendix A) and nucleic acids were stained for 10 min at room temperature with DAPI (500 nM from 14.3 mM stock; 10 min RT). Coverslips were mounted with Mowiol mounting medium, and images were acquired with the Zeiss AxioImager M2 fluorescence upright microscope equipped with ApoTome 2 module (Carl Zeiss, Inc., Oberkochen, Germany). For confocal images, immunostained samples were imaged using a 63× Plan-Apotome oil immersion objective (1.46 NA, Zeiss) in a Zeiss LSM 170 inverted microscope coupled with Airyscan detector. For stimulated emission depletion (STED) microscopy, image acquisition was performed using a 100× Plan-Apotome oil immersion objective (1.46 NA, Zeiss) in an Abberior 2D-STEDYCON module on an upright Zeiss Axio Observer Z1 confocal microscope.

### 4.11. Video Microscopy

Time-lapse video microscopy was conducted as previously described [5]. Briefly, HFFs loaded with tachyzoites were plated on 18 mm diameter coverslips coated with poly-L-Lysine (PLL 150-300 kDa, Sigma-Aldrich) and inserted in a Chamlide chambers (LCI Corp., Seoul, Korea). This magnetic cell culture chamber where temperature and CO_2_ (37 °C, 5%) are tightly controlled, was installed on an Eclipse Ti inverted confocal microscope (Nikon France Instruments, Champigny sur Marne, France) coupled to a CMOS Prime camera (Photometrics, Tucson, AZ, USA) and a CSU X1 spinning disk (Yokogawa, Roper Scientific, Lisses, France). Images were captured with a 60× Plan-neofluar objective (NA: 1.46).

### 4.12. Tachyzoite Egress and Motility Assays

HFFs were seeded on 18 mm diameter glass coverslips precoated with poly-L-lysine (150–300 kDa, 50 μg/mL in PBS, Sigma) 24 h prior to assay. They were infected with active extracellular tachyzoites in a 1 h-time window before most if not all extracellular parasites were removed with energic cold PBS washes. Cells were cultured for 40 h at 37 °C, 5% CO_2_, and then placed on the appropriate Chamlide chamber (Live Cell Instrument) in the confocal microscope under similar conditions. Intravacuolar parasite growth, parasite egress, and subsequent motility were monitored and recorded at 1.07451 s/ frame.

### 4.13. Expansion Microscopy

Expansion microscopy (proExM) was achieved according to the protocol described in [56]. Briefly, samples were prepared on 12 mm diameter glass coverslip, PFA fixed, TX-100 permeabilized and subsequently immunolabeled (Appendix A). Next, they were subjected to 0.1 mg/mL (6-((acryloyl) amino) hexanoic acid, succinimidyl ester) -Acryloyl X- diluted in PBS for 2 h at room temperature. Gelation (1 h) was completed on parafilm with the sample facing down before a 1 h incubation period in a humidified oven at 37 °C. The gel was immersed for 3 h at room temperature in digestion buffer containing 8 units/ml of proteinase K (Biolabs) and subsequently rinsed twice in PBS. A piece of digested gel was cut and expanded in a Petri dish filled with dH_2_O. During the first bath, nuclear staining with DAPI (1 µg/mL, Sigma) was performed. The gel was immobilized on a poly-L-lysine coated coverslip prior to image acquisition with either a Zeiss LSM 170 inverted microscope (Airyscan module) or a spinning disk coupled Nikon EclipseTi inverted microscope (see Section 4.11).

### 4.14. Scanning Electron Microscopy (SEM)

HFF monolayers were infected with RH_BCC7mAID x3HA_MyoJ x2Ty Tir1 tachyzoites and incubated with either 500 µM IAA or vehicle for 24 h under routine culture conditions. Freshly egressed tachyzoites were collected, filtered through a 5 µm pore size polycarbonate membrane filter (Cytiva Whatman Cyclopore 7062-2513, Fisher Scientific), and deposited on a fibronectin-pre-coated glass coverslip for 15 min. The samples were fixed with 2.5% high grade glutaraldehyde (Electron Microscopy Sciences) diluted immediately before use in 0.1 M cacodylate buffer (pH 7.2, Electron Microscopy Sciences) for 1 h at 4 °C. Samples were subsequently washed in 0.1M cacodylate buffer and left overnight at 4 °C before being transferred in PBS. Samples were ethanol dehydrated and critical point dried under CO_2_ atmosphere using a Leica EM CPD 300 apparatus prior to be coated with a 5 nm fine-grained and conductive platinum layer using the Leica high vacuum sputter coater EM ACE600. Images were captured with the Scanning Electron Microscope SU3500 (Hitachi, Tokyo, Japan). Digital images were recorded and further processed as described below.

### 4.15. Image Processing

All samples processed were randomly selected by researchers that were not blinded during the study. MetaMorph [57], ImageJ [58] and Icy [59] software were used to process the raw images from x,y,z,t stacks for events of interest. Photocompositions were realized with ImageJ and Photoshop software. 3D reconstructions were performed using 3D Slicer [60]. All images shown in the manuscript are representative of multiple independent biological replicates.

### 4.16. Statistical Analysis

Data were statistically analyzed and plots were made using GraphPad Prism 8.0 software for Windows (La Jolla, CA, USA). Data are presented as mean ± standard deviation if not indicated otherwise. Figure legends include the statistical test and resulting comparison when appropriate. For Figure 3c, an unpaired *t*-test followed by Holm–Sidak comparisons was used, with significance being represented as a *p*-value < 0.05, and the n indicated represents the parasite sample size.

## Figures and Tables

**Figure 1 ijms-23-05995-f001:**
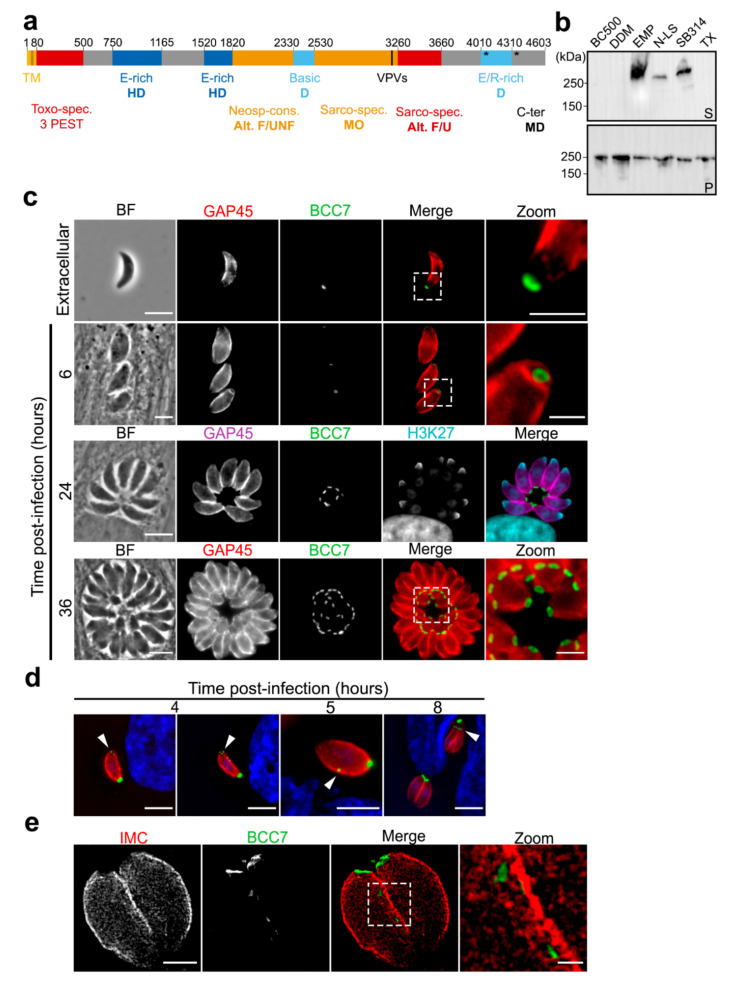
Main features of the *T. gondii* BCC7 protein. (**a**) Schematic representation of the BCC7 primary sequence based on predictive analysis. Thirteen regions were discriminated. TM, predicted transmembrane helix (residues 35–57); PEST, proline (P), glutamic acid (E), serine (S), and threonine (T) residues. In blue, regions with biased composition; in red and orange, regions conserved in *Toxoplasma* or other *Sarcocystidae* and which could be partially folded; in grey, other regions. Toxo-spec: *T. gondii* specific domain; Neosp-cons.: Neospora conserved domain; Sarco-spec: Sarcocystidae specific domain. Alt. F/UNF, Alternated Folded/UNFolded; D, Disordered; HD, highly disordered; MO, mainly ordered domains. Glutamic acid represents 26.8% of the amino-acids in the (750–1165) E-rich region and 26.3% in the (1520–1820) region. Arginine and lysine together account for 23.8% of the amino-acids of the (2330–2530) basic region while arginine and glutamic acid account for 26.6% of the E/R-rich region. The sarco-specific region includes an aromatic-rich region, and the black bar denotes 2 VPV (3340–3342 and 3348–3350) (V: valine, P: proline) repeats. * denotes ubiquitin sites at positions 4048, 4339, 4461 and 4507. (**b**) Analysis of the BCC7-HA-FLAG solubility from transgenic tachyzoites under different detergent conditions. BC500: control lysis buffer containing NP40 (0.05% *v*/*v*); DDM: BC500 supplemented with n-Dodecyl-β-D-maltopyranoside (0.1% *w*/*v*); EMP: Empigen BB (1% *v*/*v*); N-LS: N-lauryl sarcoside (0.1% *w*/*v*); SB314: sulfobetaine 3–14 (n-Tetradecyl-N,N-dimethyl-3-ammonio-1-propanesulfonate) (1% *w*/*v*); TX: Triton X-100 (0.1% *v*/*v*). Soluble (S, upper panel) and insoluble fractions (P, lower panel) from each lysate were resolved on SDS-PAGE and analyzed by Western blot using monoclonal rabbit anti-HA antibodies. The molecular weights are indicated in kiloDaltons (kDa). This figure is representative of two distinct experiments. (**c**) In situ fluorescence-based detection of BCC7-mEm in transgenic extracellular and intracellular dividing tachyzoites reveals a ring-like organization at the basal pole. The glideosome protein GAP45 is immunostained (red or purple) to visualize the parasite periphery; the apical pole and nucleus were visualized by immunostaining of tri-methylated H3K27 residues (blue) that mark the nucleus and apical cap of the parasites [24]. Scale bars: 5 µm; for zoomed areas: 2 µm. (**d**) In situ fluorescence-based detection of BCC7-mEm highlights the apical pools (4 h post invasion), lateral peripheral pools (5 h post invasion) and the enrichment at the growing end of daughter cells together with the basal pool of the mother cell (white arrowhead). The tachyzoite GAP45 is immunolabeled (red). Scale bar: 5 µm. (**e**) STED imaging of transgenic tachyzoites co-expressing BCC7-mEm and IMC-mC after immunostaining with anti-GFP antibodies (secondary antibodies conjugated to STAR Orange, green), anti-mC antibodies (secondary antibodies conjugated to STAR Red, red). Note the BCC7-mEm small pools along the IMC network and at the ring at the basal pole of dividing parasites. Appendix A corresponds to the 3D reconstruction of this sequence. Scale bar: 2 µm; for zoomed area: 500 nm.

**Figure 2 ijms-23-05995-f002:**
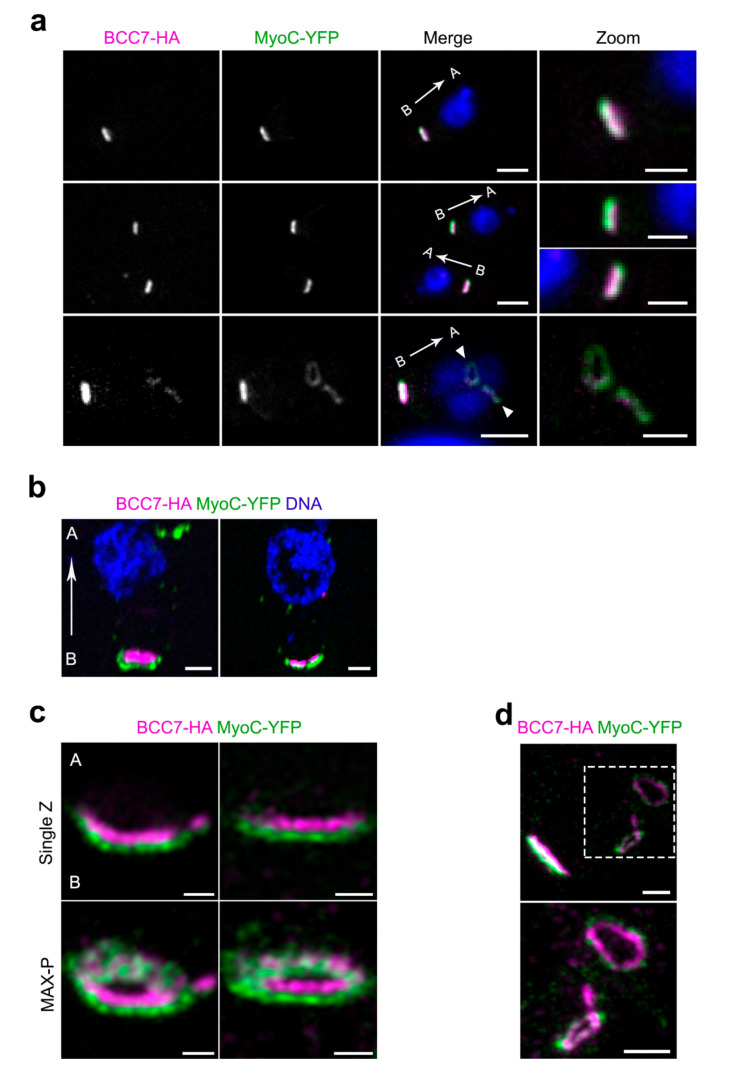
Localization of BCC7 with respect of the actomyosin motor MyoC. (**a**) HFF monolayers were infected with tachyzoites expressing BCC7-HA (magenta) and MyoC-YFP (green) tagged proteins. Following immuno-(anti-HA) and DAPI (blue) staining, image were acquired (Zeiss ApoTome.2 microscope). At this resolution, the signal from each proteins quasi overlap. White arrows show apicobasal orientation. White arrowheads mark the progeny (bottom panel). A: apex, B: base. Scale bar: 2 µm, zoom 1 µm. (**b**) Extracellular parasites expressing BCC7-HA (magenta) and MyoC-YFP (green) tagged proteins. Cells were immunostained and expanded using proExM before image acquisition (Eclipse Ti inverted confocal microscope). Note that BCC7 and MyoC can be distinguished from each other. White arrow shows apicobasal orientation. A: apex, B: base. Scale bar: 0.2 µm. (**c**) HFF monolayers were infected with tachyzoites expressing BCC7-HA (magenta) and MyoC-YFP (green) tagged proteins. Images from proExM-treated samples were acquired using Zeiss AxioImager M2 with Airyscan detector. Note the gain in resolution for the relative localization of MyoC and BCC7 in the basal pole. A: apex; B: base. Maximal projections (MAX-P) are shown in the bottom panels. Scale bar: 0.2 µm. (**d**) Dividing tachyzoites in samples from (**c**) show presence of MyoC and BCC7 ring-shaped signals in the budding daughter cells. Zoom is shown in bottom panel. Scale bars: 0.5 µm.

**Figure 3 ijms-23-05995-f003:**
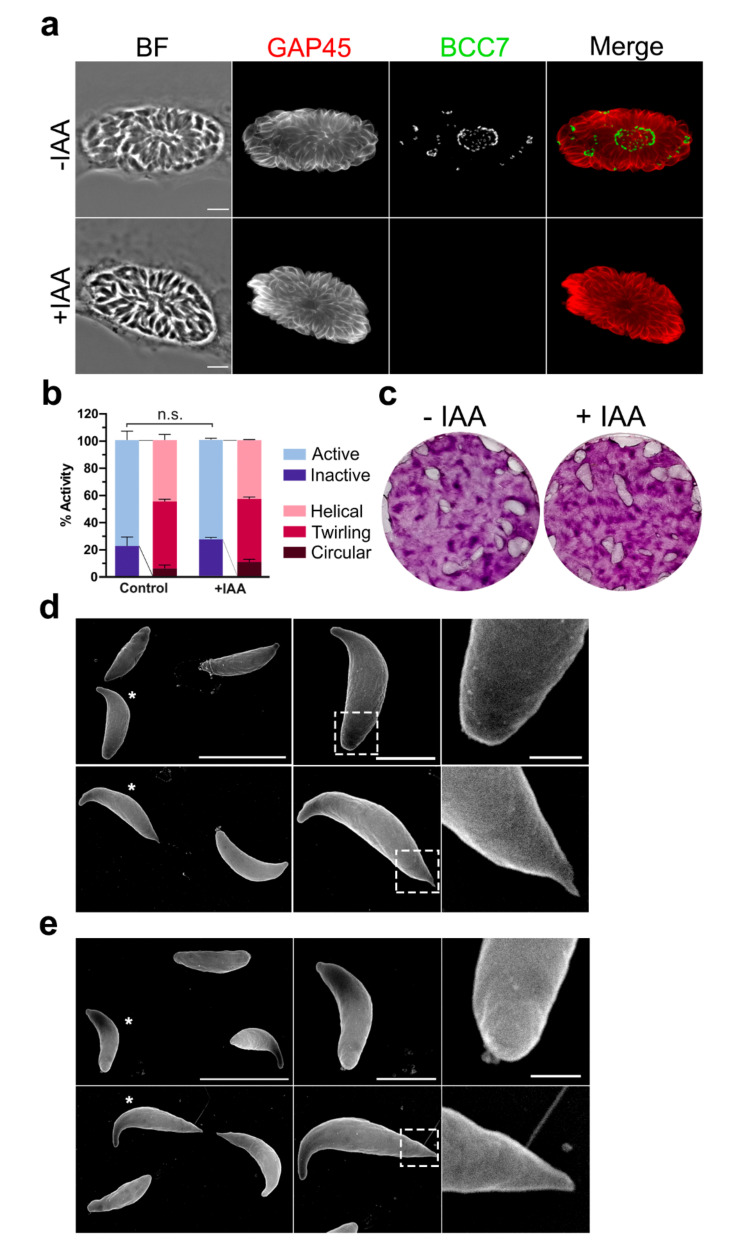
BCC7 is dispensable for the tachyzoite lytic cycle in vitro and does not impact on the ultrastructural organization of the basal pole surface. (**a**) RH_BCC7-mAID-3HA/MyoJ-2TY parasites grown in HFF cells for 48 h ± 500 µM IAA and fixed just before egress (see Appendix A). BCC7 depletion was evidenced by HA-immunostaining. No differences in the rate of progeny growing or organization as assessed by GAP45 immunostaining (red), and nuclei DAPI stain in BCC7 wild-type and BCC7-depleted parasites. Scale bar: 5 µm. (**b**) Graph showing no significant differences in the percentage of active and/or gliding BCC7 wild-type and depleted tachyzoite upon egress from the vacuole. Quantification was performed on the Appendix A (mean ± Standard Error of the Mean (SEM), unpaired *t*-test followed by Holm–Sidak comparisons method, n = 166–313 parasites from 2–3 separate experiments, n.s.: non-significant, *p*-value = 0.622399). (**c**) Plaque formation in HFF monolayers over a 7-day infection. Relative size of the plaques showed no significant differences between BCC7 wild-type and BCC7 depleted parasites. These images are representatives of three different experiments. (**d**,**e**) Scanning Electron Microscopy (SEM) images of RH_BCC7-mAID-3HA. BCC7 expressing (**d**) or BCC7-depleted (**e**) extracellular tachyzoites did not show major ultrastructural differences of their basal pole regardless of their resting tachyzoites or acting status. SEM field overview on the left, zoom on the parasite selected with an asterisk (middle column), and a second zoom on the parasite’s basal pole (pointed square, right column). Scale bars: left: 5 µm; middle: 2 µm; right: 500 nm.

**Figure 4 ijms-23-05995-f004:**
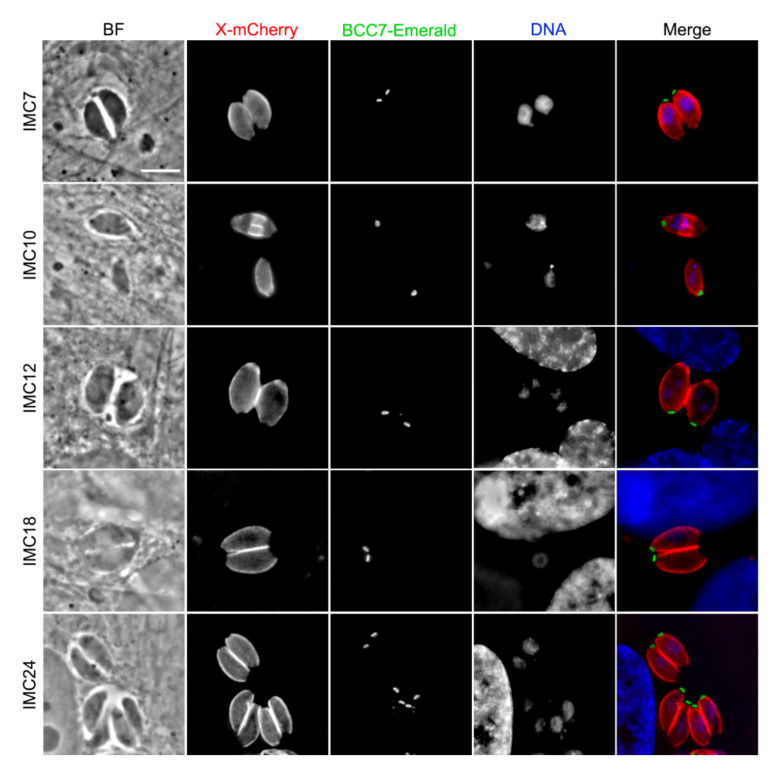
Images of IMC proteins identified in this study as putative BCC7-binding proteins. HFF monolayers were infected with freshly egressed parasites expressing endogenously fused BCC7-mEm (green) and IMC-mC tagged proteins (red). Samples were fixed and stained with DAPI (blue) before image acquisition (ZEISS ApoTome.2 microscope). The elongating IMC from the budding progeny contains IMC10. Scale bar, 5 µm.

**Figure 5 ijms-23-05995-f005:**
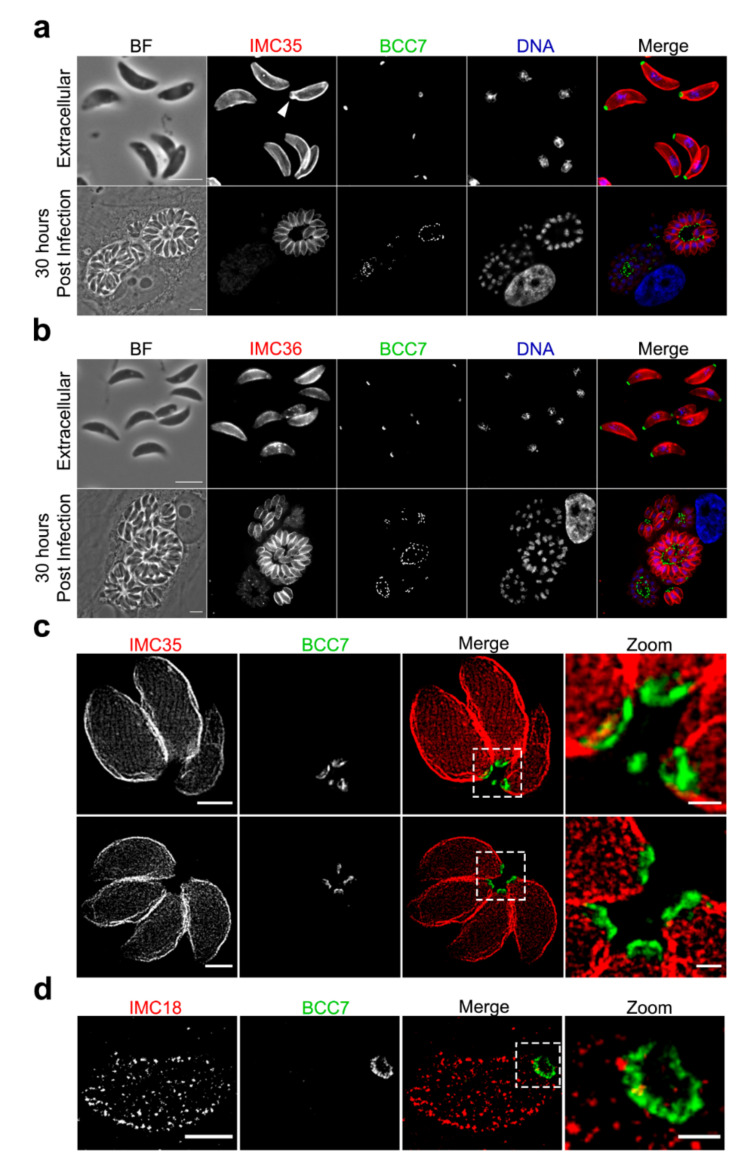
Cell-cycle-dependent expression of IMC35 and IMC36. Parasites expressing endogenously fused BCC7-mEm (green) and IMC35-mC (**a**) or IMC36-mC (**b**) proteins (red) were fixed and stained with DAPI (blue) before images acquisition (ZEISS ApoTome.2 microscope). Different steps of the division process were observed as indicated. IMC35 showed a continuity with the BCC7 ring structure on extracellular parasites (white arrowhead). Scale bars: 5 µm. (**c**) STED imaging of tachyzoites co-expressing BCC7-mEm and IMC35-mC and immunostained with anti-GFP and anti-mCherry antibodies respectively. Appropriate secondary antibodies conjugated to either Abberior STAR Orange (BCC7-mEm, green) or Abberior STAR Red (IMC35-mC, red) were used. Note the pattern alignment of IMC35 with the microtubule basket orientation and the spatial proximity between IMC35-mC and BCC7-mEm at the basal pole. (**d**) STED imaging of a rosette of intracellular parasites expressing endogenously fused BCC7-mEm and IMC36-mC using the same staining as in (**c**). Scale bars for (**c**,**d**): 2 µm; for zoomed areas: 500 nm.

**Table 1 ijms-23-05995-t001:** List of putative BCC7-binding proteins identified by co-IP coupled to MS-based proteomics. Identity of proteins was assigned by ToxoDB [19] unless specified (*) and their subcellular location was determined by Hyper Lopit approach [20]. Parasite fitness during the lytic cycle was evaluated from the genome-wide CRISPR screen [27].

Partner	TGGT1 Accession Number	Description	MW (kDa)	Proteomic Data		Subcellular Location	Phenotype Score
				Log2FC	*p*-Value		
-	311230	BCC7	494.8	8.71	2.68 × 10^−7^	unknown	**0.74**
1	222220	IMC7	46.8	3.44	1.96 × 10^−3^	IMC	**−0.64**
2	258470	IMC24 [32]	31.7	3.26	1.10 × 10^−3^	IMC	**2.12**
3	230940	hypothetical protein	155.2	2.95	7.63 × 10^−3^	unknown	**−4.92**
4	233450	SAG-related sequence SRS29A	44.2	2.71	4.61 × 10^−4^	PM-peripheral 1	**2.42**
5	248700	IMC12	29.8	2.38	6.88 × 10^−3^	IMC	**−0.17**
6	212260	TgZFP2 [33]	38	2.47	4.11 × 10^−2^	unknown	**−4.5**
7	216670	FUSE-binding protein 2	100	2.19	4.86 × 10^−2^	nucleus-non chromatin	**−3.31**
8	230160	hypothetical protein	15.8	2.19	1.54 × 10^−2^	IMC	**2.11**
9	236950	hypothetical protein	12.5	2.13	1.07 × 10^−2^	IMC	**0.96**
10	230210	IMC10	61.2	2.05	1.19 × 10^−2^	IMC	**−4.7**
11	295360	IMC18 [32]	29.6	2.00	1.45 × 10^−2^	IMC	**1.13**
12	315610	hypothetical protein	15.9	2.67	8.17 × 10^−4^	unknown	**1.33**
13	231160	hypothetical protein	18.3	2.41	3.11 × 10^−3^	IMC	**−1.6**

## Data Availability

The mass spectrometry proteomics data have been deposited to the ProteomeXchange Consortium via the PRIDE (PMID: 34723319) partner repository with the dataset identifier PXD032362.

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
