# Peer review of "The BCC7 Protein Contributes to the Toxoplasma Basal Pole by Interfacing between the MyoC Motor and the IMC Membrane Network"

_ijms, 2022, doi:10.3390/ijms23115995_

Round 1
Reviewer 1 Report
The authors described an extensive study of the BCC7 protein which is component of Toxoplasma gondii basal complex to characterize this protein with respect to the posterior myosin J and myosin C motors. This study is well planned and well conducted using several techniques like proteomics and microscopic methods. The manuscript is clearly organized and enhanced with valuable supplementary materials. I am recommending acceptance of the manuscript after one small correction as described below.
I have only one minor point that should be corrected:
- Line 587: “Predicted phosophorylation sites were retrieved from [46].”. The second part of the sentence is missing, please correct this.
Author Response
We thank the reviewer since this point deserved to be clarified. It now reads in the text:
Ordered and disordered regions have been predicted using the FoldIndex webserver [50]. Predicted phosphorylation sites were retrieved from ToxoDB documented based on the phosphoproteomic studies on Toxoplasma and Plasmodium zoites [51].
Accordingly, the reference has been added as # 51: “The phosphoproteomes of Plasmodium falciparum and Toxoplasma gondii reveal unusual adaptations within and beyond the parasites' boundaries. Treeck et al. Cell Host Microbe 2011 Oct 20;10(4):410-9 PMID: 22018241, DOI: 10.1016/j.chom.2011.09.004
Reviewer 2 Report
This article shows a fine characterization of the subcellular localization of BCC7 protein, a non-essential basal pole marker of Toxoplasma gondii. BCC7 forms a ribbed ring with MyoC just below, there is also evidence supporting an apicobasal trafficking of this protein. Despite its non-essentiality during the in vitro lytic cycle (cell lines), it may be essential in vivo (organism) and/or during the other phase of the life cycle of this parasite. Furthermore, the authors identified and characterized 13 putative BCC7 partners, being part of them members of the Inner Membrane Complex (IMC) family and 2 of them described for the first time as IMC members.
The article is clear, relevant, well-structured and the experimental design was appropriated.
The results, figures and tables are well presented and are easy to follow and interpreted.
Below are minor revisions:
- Lines 108 and 109: between these 2 paragraphs is missing “connection”. The reader is following a basal complex history and, suddenly, HDAC appears. As a suggestion, maybe you can start by the end, making the connection through the basal complex BioID and then telling the past of the BCC7 protein.
- Line 157: please remove “in native state”. You are using detergents to solubilize the protein and analyzing the protein extracts in SDS-PAGE, all denaturing conditions.
- Figure 1a: the scheme is too dense with a lot of information; it is difficult to analyze. As a suggestion, I would remove information from the scheme and pass to the legend.
- Figure 1b: in the insoluble fraction BCC7 looks to have around 250kDa. In the soluble fraction, it is a little bit above 250kDa but, despite not having a larger marker, doesn’t look like around 500kDa. This fact should be commented on in the text.
- Line 176: what do you mean with “synchronously infected”? Was it because you washed/removed the free parasites? Maybe it is better to clarify in the text.
- Lines 302-303: Did you try to test the efficient depletion of BCC7 also by western blot?
- Line 418: remove “data not shown”. If it is relevant, add it to supplemental material. Otherwise, it is better to remove.
- Line 434: check the T. gondii “italic”.
- Figure 5c: Did you try to do an IF with IMC33-mc and acetylated tubulin (sigma antibody), to better characterize the colocalization with the subpellicular microtubules?
- Line 655: remove one “in” from “stacked in in”
- Line 847: I was only able to find the reference 12 in biorxiv, if it is true, you should cite from there (doi:https://doi.org/10.1101/2021.10.14.464364)
- Line 850: reference 13 is from Front. Cell. Infect. (https://doi.org/10.3389/fcimb.2022.882166).
In this reference it is described a BCC7 KO, with no significant reduction in plaque forming but with a less incidence of parasite organization in rosettes suggesting a role in the formation and/or maintenance of the cytoplasmic bridge. You should discuss/comment in comparison with your results.
Author Response
Lines 108 and 109: between these 2 paragraphs is missing “connection”. The reader is following a basal complex history and, suddenly, HDAC appears. As a suggestion, maybe you can start by the end, making the connection through the basal complex BioID and then telling the past of the BCC7 protein.
We have followed the advice of the reviewer and rephrased the paragraph to provide more fluidity in the reading; It now reads:
“A proteomic analysis carried on a detergent-extracted subpellicular cytoskeleton fraction allowed identifying a 500 kDa protein encoded by TGGT1_311230 [17] which was subsequently selected in a Yeast two-hybrid (Y2H) screen with MORN1 as a bait [18] therefore possibly involved in the shaping of the basal pole. We also identified this protein in a screen for Toxoplasma tachyzoite Histone Deacetylase 3 (HDAC3) substrate candidates and further invalidated its HDAC3 substrate potential, but we noticed the striking posterior localization of an epitope-tagged version of the protein. This in situ observation prompted us to map the protein position with respect to the posteriorly located myosins. In the course of our study, a BioID-based map of the basal complex (BC) proteome also picked up this protein as a basal complex component and named it as BCC7 [12,13]. Using a combination of biochemistry, co-immunoprecipitation coupled to mass spectrometry (MS)-based proteomics, live and super resolution laser scanning microscopy, we now bring evidence for BCC7 positioned at the interface between a set of IMC proteins including two newly identified herein, and the MyoC motor”.
Line 157: please remove “in native state”. You are using detergents to solubilize the protein and analyzing the protein extracts in SDS-PAGE, all denaturing conditions.
We have removed the “native state”.
Figure 1a: the scheme is too dense with a lot of information; it is difficult to analyze. As a suggestion, I would remove information from the scheme and pass to the legend.
We have removed part of the information carried on the panel A Figure 1 which is now available in the legend as follows,
“…. (a) Schematic representation of the BCC7 primary sequence based on predictive analysis. Thirteen regions were discriminated. TM, predicted transmembrane helix (residues 35-57); PEST, proline (P), glutamic acid (E), serine (S), and threonine (T) residues. In blue, regions with biased composition; in red and orange, regions conserved in Toxoplasma or other Sarcocystidae and which could be partially folded; in grey, other regions. Toxo-spec: T. gondii specific domain; Neosp-cons.: Neospora conserved domain; Sarco-spec: Sarcocystidae specific domain. Alt. F/UNF, Alternated Folded/UNFolded; D, Disordered; HD, Highly Disordered; MO, Mainly Ordered domains. Glutamic acid represents 26.8% of the amino-acids in the (750-1165) E-rich region and 26.3% in the (1520-1820) region. Arginine and lysine together account for 23.8% of the amino-acids of the (2330-2530) basic region while arginine and glutamic acid account for 26.6% of the E/R-rich region. The sarco-specific region includes an aromatic-rich region, and the black bar denotes 2 VPV (3340-3342 and 3348-3350) (V: valine, P: proline) repeats. * denotes ubiquitin sites at positions 4048, 4339-4461-4507.”
Figure 1b: in the insoluble fraction BCC7 looks to have around 250kDa. In the soluble fraction, it is a little bit above 250kDa but, despite not having a larger marker, doesn’t look like around 500kDa. This fact should be commented on in the text.
Migration and transfer of such a high molecular weight protein is challenging and the resolution of protein bands >250 kDa in SDS-PAGE conditions is usually low. Despite particular attention to sample preparation and gel migration conditions, some variations can be observed. However, when run on the same gel, the size of either insoluble or soluble BBC7 pools were found similar (Fig S4). This information is now added to the text, section results first paragraph:
“Although the migration in SDS-PAGE 4-12% gradient gels of BCC7 did not seem to depend on the detergent used and the solubility or insolubility status (Figure S4), we observed some variability in the migration pattern between experiments with products migrating faster but only one band was always observed arguing for a full-length product”.
Line 176: what do you mean with “synchronously infected”? Was it because you washed/removed the free parasites? Maybe it is better to clarify in the text.
We agree with this remark. Indeed, it is not a formal synchronization of parasite with respect to their cell cycle prior to invasion. We are not necessarily starting with ~100% “mature” parasites (post the short cytokinesis event) because of the intrinsic variability in the cell cycle of the progeny inside each of the host cell. Here we refer to a well define and rather short period during which we promote parasite-host cell contact for invasion, to avoid multiplying the variability in starting intracellular replication. To this end, we removed as much as possible of the noninternalized parasites by extensive washing of the host cell monolayer. Therefore, we removed the term “synchronously” and rephrased the text as follows:
« Following a 30 min period of contact between freshly egressed tachyzoites and HFF monolayers, the non-internalized parasites were removed and the development of the internalized ones was further monitored ».
Lines 302-303: Did you try to test the efficient depletion of BCC7 also by western blot?
In parallel assays, we have used another RHΔhxgprtΔku80 line of T. gondii that stably express TIR1-Myc and was provided by MA Hakimi. We have introduced the mini-AID-BCC7-3HA construct at the endogenous locus and controlled by Western blot the significant decrease of the BCC7 protein level that was in line with almost no labeling of CRC7-HA when observed in IFA using anti -HA antibodies. In the particular set of assays presented in the manuscript, we have used the RHΔhxgprtΔku80 expressing TIR1-Flag(x3) provided by D. Sibley in which the mini-AID-BCC7-3HA construct was introduced using CRISPR/Cas9 genome editing, as presented Figure S3 panels A and B. In these cases, we could not detect any fluorescent signal upon 40 h of Auxin treatment of the infected cells when performing the same IFA assays with anti-HA antibodies, and we have not made the corresponding Western control. If these blots are judged necessary as additional information of protein depletion, we will de-freeze the parasite line and carry the Western blot assays.
Line 418: remove “data not shown”. If it is relevant, add it to supplemental material. Otherwise, it is better to remove.
We agree with this remark and it was in fact left from a previous draft where the data was not introduced; We have in the revised version removed the “data not shown” statement. The nuclear localization is visible on the Figure S5, bottom panel together with other putative BCC7 partners.
Line 434: check the T. gondii “italic”.
Done
Figure 5c: Did you try to do an IF with IMC33-mc and acetylated tubulin (sigma antibody), to better characterize the colocalization with the subpellicular microtubules?
We thank the reviewer for this relevant suggestion that we intend to follow since we routinely use microtubule labelling of tachyzoites including using the acetylated tubulin antibodies. We will also be happy to offer the images if nice enough for the cover suggestion of the Issues.
Line 655: remove one “in” from “stacked in in”
Done
Line 847: I was only able to find the reference 12 in biorxiv, if it is true, you should cite from there (doi:https://doi.org/10.1101 /2021.10.14.464364)
The reviewer is correct and we have added the citing accordingly
Line 850: reference 13 is from Front. Cell. Infect. (https://doi.org /10.3389/fcimb.2022.882166).
In this reference it is described a BCC7 KO, with no significant reduction in plaque forming but with a less incidence of parasite organization in rosettes suggesting a role in the formation and/or maintenance of the cytoplasmic bridge. You should discuss/comment in comparison with your results.
We have rephrased the text so to provide some comparative information. It now reads:
“However, that BCC7 was found dispensable in vitro for T. gondii tachyzoite perpetuation based on plaque formation capacity was surprising. A detailed study by Gubbels et al. published in the course of our manuscript evaluation [12] points the subtle effect of BCC7 loss following gene knock out strategy on the rosette organization of the progeny within vacuole which appears to be less consistent than in wild type, and which do not impact on the general in vitro fitness score. This defect suggests that BCC7 could assist in the formation and maintenance of the cytoplasmic bridge between progeny after division within the parasitophorous vacuole which would translate into a rosette disorganization. Of note, other BC components identified in the Gubbels et al. study were found to separately affect the parasite alignment in rosette to various extents, indicating that T. gondii has likely evolved a versatile multi-component process that introduces molecular plasticity in the shaping of mature cell upon the final step of division.”
